# Case fatality rate and its determinants among admitted stroke patients in public referral hospitals, Northwest, Ethiopia: A prospective cohort study

Gashaw Walle Ayehu[1]*, Getachew Yideg Yitbarek[1], Tadeg Jemere[1], Ermias Sisay Chanie[2], Dejen Getaneh Feleke[2], Sofonias Abebaw[3], Edgeit Zewde[1], Daniel Atlaw[4], Assefa Agegnehu[1], Ayele Mamo[5], Sisay Degno[6], Melkalem Mamuye Azanaw[3]

1 Department of Biomedical Sciences, College of Health Sciences, Debre Tabor University, Debre Tabor, Ethiopia, 2 Department of Pediatrics and Child Health Nursing, College of Health Sciences, Debre Tabor University, Debre Tabor, Ethiopia, 3 Department of Public Health, College of Health Sciences, Debre Tabor University, Debre Tabor, Ethiopia, 4 Department of Human Anatomy, Madda Walabu University, Goba, Oromia, Ethiopia, 5 Department of Pharmacy, School of Medicine, Goba Referral Hospital, Madda Walabu University, Bale-Goba, Ethiopia, 6 Department of Public Health, Shashemene Campus, Madda Walabu University, Shashemene, Ethiopia

* gashawwalle01@gmail.com

**Data Availability Statement:** All relevant data are within the paper and its Supporting Information files.

## Abstract

According to the global burden of disease 5.5 million deaths were attributable to stroke. The stroke mortality rate is estimated to be seven times higher in low-income countries compared to high-income countries. The main aim of the study was to assess the 28 days case fatality rate and its determinants among admitted stroke patients in public referral hospitals, in Northwest Ethiopia. A hospital-based prospective cohort study was conducted from December 2020 to June 2021. The study population was 554 stroke patients. Based on Akakian Information Criteria, the Gompertz model was fitted to predict the hazard of death. The study included admitted stroke patients who were treated in the general medical ward and neurology ward. The mean age of the participants was 61 ± 12.85 years and 53.25% of the patients were female. The 28-days case fatality rate of stroke was 27.08%. The results from Gompertz parametric baseline hazard distribution revealed that female sex adjusted hazard rate (AHR = 0.27, 95% CI:0.18–0.42), absence of a family history of chronic disease (AHR = 0.50, 95%CI:0.29–0.87), good GCS score (AHR = 0.21, 95% CI:0.09–0.50) and the absence of complication during hospital admission (AHR = 0.16, 95% CI:0.08–0.29) were factors which decrease hazard of 28 days case fatality rate. While, hemorrhagic stroke subtype (AHR = 1.38, 95% CI:1.04–3.19), time from symptom onset to hospital arrival (AHR = 1.49, 95% CI:1.57–2 .71), time from confirmation of the diagnosis to initiation of treatment (AHR = 1.03, 95% CI:1.01–1.04), a respiratory rate greater than 20 (AHR = 7.21, 95% CI:3.48–14.9), and increase in NIHSS score (AHR = 1.16, 95% CI:1.10–1.23) were factors increasing hazard of 28 days case fatality rate of stroke. At 28-days follow-up, more than one-fourth of the patients have died. The establishment of separate stroke centers and a

**Funding:** This study was conducted by funding obtained from Debre Tabor University but the funding institution has no role in the design, collection, analysis, interpretation of the result, and writing the manuscript.

**Competing interests:** The authors have declared that no competing interests exist.

network of local and regional stroke centers with expertise in early stroke evaluation and management may address challenges.

## Introduction

In 2020, stroke was turned into the second leading cause of death after ischemic heart disease [1]. WHO estimated that globally 15 million people suffer from stroke each year [2]. Global death due to ischemic stroke is lower than due to hemorrhagic stroke [1]. Stroke mortality is estimated to be seven times higher in low-income countries compared to high-income countries [2]. Sub- Saharan Africa (SSA) is undergoing an epidemiologic transition where the increasing morbidity and mortality of non-communicable disease is competing with the already surging burden of infectious disease [3].

A systematic analysis for the global burden of disease reported, globally the age-standardized rate of deaths due to stroke decreased by 36·2% from 1990 to 2016, these death rates also declined for all but southern sub-Saharan Africa have no significant change in death rate [1].

The best method to control the stroke burden and meet the global goal of a 2% reduction each year is by primary prevention via early detection of stroke risk factors. A study showed that 90.5% of the global burden of stroke was attributable to modifiable factors [4]. Unlike in developing countries, stroke mortality is decreasing in the developed world [5]. The decreased percentage of stroke hospitalization and mortality in developed countries likely reflects the advancements in acute stroke care [6].

On the contrary, the evidence shows an increase in trend and mortality of stroke in SSA, good-quality data on mortality of stroke patients in developing countries specifically Ethiopia are still deficient. So, this study aimed to assess the 28-days case fatality rate and determinants of case fatality rate among admitted stroke patients in public referral hospitals, in Northwest Ethiopia, from December 2020 to June 2021.

## Methods

### Study design, and settings

A prospective cohort study design was conducted in three public referral hospitals located in Northwest Ethiopia, including the University of Gondar teaching hospital located in Gondar, Tibebe Gion comprehensive specialized hospital, and Felege Hiwot referral hospital which are located in Bahr Dar. All these hospitals are the most equipped in northwest Ethiopia specifically in having CT scans as an imaging modality. The study period was 7 months from December 2020 to June 2021.

### Study population

The source population was all stroke patients admitted to the public referral hospitals in Northwest Ethiopia. The study population was all stroke patients diagnosed and confirmed during the study period in the three referral hospitals. Five hundred fifty-four adult stroke patients were included.

### Inclusion and exclusion criteria

All patients (age ≥ 18 years) diagnosed with stroke and admitted to the medical or neurology ward of the three public hospitals during the study period were included. The exclusion criteria were specified to exclude admitted patients who died before confirmation of diagnosis, after further evaluation, if the initial assessment or diagnosis of stroke was later changed to

another case (ruled out stroke) and patients who were being treated by other health institutions, referred after treatment or due to complications.

## Study variables

**Dependent variable.** Twenty-eight days case fatality rate of admitted stroke patients.

**Independent variables.** Socio-demographic and behavioral characteristics (age, sex, residence, marital status, occupation, educational level, alcohol use, smoking habit, diet habit, and activity level). Patient characteristics (clinical presentation, presence of co-morbidities and complications, time from onset of symptom to hospital arrival, time to CT scan and time start of treatment from confirmation of diagnosis, type of stroke, length of hospital stay, GCS score, mRS score NIHSS score, respiratory rate, temperature, pulse rate blood pressure, and ICH score) were the independent variables of the study.

## Operational definition

**Event**: death recorded among admitted stroke patients during 28 days of follow-up (in the hospital or after hospital discharge in the period reported by caregiver).

**Censored**:(i) if the patient is alive until the 28th day of admission, (ii) if the patient is dead before the diagnosis is confirmed, (iii) if the cause of death is not a stroke, and (iv) if the patient is referred to other health institutes.

**Time to event**: Time from hospital admission until the death of the patient is confirmed either during admission or after discharge within the 28 days of follow-up.

**Twenty-eight days case fatality rate**: was calculated using the number of deaths due to stroke during the 28 days follow-up period as the numerator and the total number of stroke patients during the 28-day follow-up period as the denominator.

## Data collection procedure

Data collection was carried out by one internal medicine resident (R-3) for each hospital with training on the contents of the data collection tool. Data collectors collected all relevant data from patients' charts and interviewed the patients/caregivers using a prepared data extraction form and questionnaire. History used for the study was taken from the patient and/or relatives in the language they understood (in Amharic). Important data on the investigations done, medication, duration of treatment, hospital arrival date was collected by chart review, Controversial data like laboratory interpretation, drug therapy problems, and unclear data on patients' charts were discussed with physicians working in the medical/ neurology ward of the Hospitals during the study period. Data on the event (mortality) was collected as the event happens if it is in the hospital or if the event happens after discharge it was collected by close telephone follow up every week until the 28th day from admission. Using interview baseline socio-demographic information, clinical presentation, past medical illness, duration of symptom, time from symptom onset to hospital arrival, from hospital arrival to CT-scan, time from hospital arrival to treatment initiation was collected by interview while, findings of imaging's like chest X-rays, and brain imaging encountered complications and management approaches of stroke patients, death during admission was recorded from the death report. Finally, the death of the patient after discharge was recorded using close telephone follow-up.

## Data quality assurance

To maintain data quality, the data collection instrument which consisted of the interview questionnaire and the data extraction format was assessed by a neurologist for clarity and

comprehensiveness of its contents. Pre-testing was done on 5% (28) of the study participants before the start of the actual study at Debre Tabor comprehensive specialized hospital. All the necessary modifications and adjustments were done before implementing in the main study. The principal investigator close supervised the data collection process. To ensure its completeness and consistency, the maximum effort was done at the level of data entry, analysis, interpretation, and presentation to maintain the quality of the data.

## Data processing and analysis

The collected data were coded, checked, and entered by using Epi-info version 7. It was cleaned and edited by simple frequencies and cross-tabulation before analysis. Then, the data was exported to STATA version 16 software for analysis. Descriptive statistics and numerical summary measures were presented using frequencies distribution tables.

Candidate variables were selected at $p < 0.25$ on binary cox regression for multivariable cox regression. On multivariable Cox regression predictors with a probability value, less than 0.05 was considered statistically significant. A global test was fitted and showed the model was inadequate. The best fit model was selected using Akakian Information Criteria (AIC) and the Log-likelihood ratio test. The lowest Akakian Information Criteria and the highest Loglikelihood ratio value indicate the best fit model, among those Gompertz model was selected. Finally, the Gompertz model was fitted to predict the hazard of death. The 28 days case fatality rate of different variables was compared using the cox-proportional hazards regression model and the Log-Rank test.

## Ethical approval and consent of the participant

The study was performed under the ethical standards of the Helsinki declaration, Ethical clearance was obtained from the Ethical Review Committee of the College of Health Science, Debre Tabor University with the reference number of CHS/3238/2013 E.C. Copy of ethical clearance was submitted to medical directors of the three hospitals. Matrons of the medical/neurology wards of the hospitals were informed. Informed written consent was obtained from each selected patient/ when the patients are not able to give informed written consent was obtained from a close relative or caregiver. Neither the case records nor the data extracted were used for any other purpose. The confidentiality and privacy of patients were assured throughout by removing identifiers from data collection tools using different codes.

## Results

### Sociodemographic, and behavioral characteristics of participants

A total of 554 participants were being treated in the medical and neurology ward. The majority 295(53.25%) of the participants were females. The mean age of the participants was 61 ± 12.85 years. The mean age of patients who died during 28 days of follow-up was 60 ±13.3 years and those who were alive were 61.29 ± 9.6 years. Ischemic stroke was the most prevalent sub-type in both sexes (male 29.60% and female 30.68%) (Table 1). The average time from symptom onset to hospital arrival was 40 hours ranging from 1-to 336 hours. while the mean duration from hospital arrival to CT scan was 2.48 ±2.28 hours. In addition, the time from the confirmation of the diagnosis to the start of treatment was 4.38 hours (SD = 9.45). The mean time from confirmation of the diagnosis to the start of treatment among patients who died was 6.36 hours and 3.64 hours for patients who were alive (Table 2).

**Table 1. Distribution of sociodemographic and behavioral factors in male and female stroke patients in public referral hospitals, Northwest Ethiopia, 2021.**

| VARIABLES | | MALE N (%) | FEMALE N (%) | TOTAL = 554 N (%) |
|---|---|---|---|---|
| AGE CATEGORY | 20–29 | 6 (1.08) | 3 (0.54) | 9 (1.62) |
| | 30–39 | 5 (0.90) | 15 (2.70) | 20 (3.61) |
| | 40–49 | 15 (2.70) | 45 (8.12) | 60 (10.83) |
| | 50–59 | 55 (9.93) | 70 (12.63) | 125 (22.56) |
| | 60–69 | 85 (15.34) | 99 (17.87) | 184 (33.21) |
| | 70–79 | 60 (10.83) | 60 (10.83) | 120 (21.66) |
| | 80–89 | 30 (5.41) | 6 (1.08) | 36 (6.5) |
| RESIDENCE | Urban | 100 (18.05) | 75 (13.54) | 175 (31.59) |
| | Rural | 159 (28.7) | 220 (39.7) | 379 (68.41) |
| MARITAL STATUS | Unmarried | 15 (2.70) | 0 (0) | 15 (2.70) |
| | Married | 229 (41.33) | 240(43.32) | 469 (84.65) |
| | Divorced | 15(2.70) | 20 (3.61) | 35 (6.32) |
| | Widowed | 0 (0) | 35 (6.32) | 35 (6.32) |
| OCCUPATION | Farmer | 160 (28.90) | 5 (0.90) | 165 (29.78) |
| | Merchant | 35 (6.32) | 10 (1.80) | 45 (8.12) |
| | Retired | 30 (5.41) | 5 (0.90) | 35 (6.32) |
| | Government employee | 34 (6.13) | 0 (0) | 34 (6.13) |
| | Housewife | 0 (0) | 275(49.64) | 275 (49.64) |
| EDUCATIONAL STATUS | No formal education | 140 (25.27) | 235(42.42) | 375 (67.69) |
| | Able to read and write | 55 (9.92) | | 85 (15.34) |
| | Primary | 15 (2.70) | 30 (5.41) | 35 (6.32) |
| | Secondary | 14 (2.53) | 20 (3.61) | 19 (3.43) |
| | Diploma | 25 (4.51) | 5 (0.90) | 25 (4.51) |
| | Degree and above | 10 (1.80) | 0 (0) | 15 (2.70) |
| | | | 5 (0.90) | |
| ALCOHOL DRINKING | Non-drinker | 120 (21.66) | 185(33.39) | 305 (55.05) |
| | Past drinker | 100 (18.05) | 90 (16.24) | 190 (34.30) |
| | Current drinker | 39 (7.04) | 20 (3.61) | 59 (10.65) |
| SMOKING HABIT | Non-smoker | 240 (43.32) | 290(52.34) | 530 (95.67) |
| | Past smoker | 19 (3.43) | 5 (0.90) | 24 (4.33) |
| FRUIT AND VEGETABLE EATING | < 2 times per week | 244 (44.04) | 285(51.44) | 529 (95.48) |
| | 3–4 times per week | 15 (2.70) | 10 (1.80) | 25 (4.51) |
| PHYSICAL ACTIVITY LEVEL | Extremely inactive | 10 (1.80) | 10 (1.80) | 20 (3.60) |
| | Sedentary | 35 (6.32) | 60 (10.83) | 95 (17.15) |
| | Moderately active | 165 (29.78) | 210(37.90) | 375 (67.69) |
| | Vigorously active | 5 (0.90) | 10 (1.80) | 15 (2.70) |
| | Extremely active | 44 (7.94) | 5 (0.90) | 49 (8.84) |
| TYPE OF STROKE | Ischemic stroke | 164 (29.60) | 170(30.68) | 334 (60.29) |
| | Hemorrhagic stroke | 95 (17.15) | 125(22.56) | 220 (39.71) |

## 28-days follow up outcome and determinants of case fatality rate

Overall, the 28-days case fatality rate was 27.08% (150), the majority of deaths were recorded among hemorrhagic stroke patients 60% (90) and the remaining 40% (60) were ischemic stroke respectively. During the 28 days follow up period, one hundred twenty stroke patients (80%) were dead during in-hospital treatment while the remaining 20% (30) were dead after they were discharged. At 28-days follow-up case fatality rate was slightly higher among male stroke patients 53.3% (80).

**Table 2.** Sociodemographic, behavioral, and baseline characteristics of stroke patients who were dead and alive after 28 days of follow up in public referral hospitals, Northwest Ethiopia.

| Variable | | Dead at 28 days N (%) | Alive at 28 days N (%) |
|---|---|---|---|
| Sex | Male | 80 (14.44) | 179 (32.31) |
| | Female | 70 (12.63) | 225 (40.61) |
| Age | 20–29 | 0 | 6 (1.1) |
| | 30–39 | 5 (0.9) | 15 (2.7) |
| | 40–49 | 20 (3.6) | 40 (7.2) |
| | 50–59 | 30 (5.4) | 95 (17.15) |
| | 60–69 | 69 (12.45) | 115 (20.75) |
| | 70–79 | 15 (2.7) | 105 (15.9) |
| | 80–89 | 11 (2.0) | 28 (5.0) |
| Residence | Urban | 46 (8.3) | 129 (23.3) |
| | Rural | 104 (18.77) | 275 (49.64) |
| Marital status | Unmarried | 0 | 15 (2.7) |
| | Married | 140 (25.27) | 329 (59.38) |
| | Divorced | 5 (0.9) | 30 (5.4) |
| | Widowed | 10 (1.8) | 25 (4.5) |
| Occupation | Farmer | 45 (8.13) | 120 (21.6) |
| | Merchant | 15 (2.7) | 30 (5.4) |
| | Retired | 15 (2.7) | 20 (3.6) |
| | Government employee | 10 (1.8) | 24 (4.33) |
| | Housewife | 65 (11.7) | 210 (37.9) |
| Educational status | No formal education | 105(18.9) | 270 (48.7) |
| | Able to read and write | 30 (5.4) | 55 (9.9) |
| | Primary | 5(0.9) | 30 (5.4) |
| | Secondary | 0 | 19 (3.4) |
| | Diploma | 10 (1.8) | 15 (2.7) |
| | Degree and above | 0 | 15 (2.7) |
| Alcohol drinking | Non-drinker | 70 (12.6) | 235 (42.4) |
| | Past drinker | 65 (11.7) | 125 (22.5) |
| | Current drinker | 15 (2.7) | 44 (7.9) |
| Smoking habit | Non-smoker | 140 (25.3) | 390 (70.4) |
| | Past smoker | 10 (1.8) | 14 (2.5) |
| Fruit and vegetable eating | < 2 times per week | 145(26.17) | 384 (69.3) |
| | 3–4 times per week | 5 (0.9) | 20 (3.6) |
| Physical activity level | Extremely inactive | 0 | 20 (3.6) |
| | Sedentary | 20 (3.6) | 75 (13.5) |
| | Moderately active | 105 (18.9) | 270 (48.7) |
| | Vigorously active | 5 (0.9) | 10 (1.8) |
| | Extremely active | 20 (3.6) | 29 (5.2) |
| Type of stroke | Ischemic stroke | 60 (10.83) | 274 (49.46) |
| | Hemorrhagic stroke | 90 (16.24) | 130 (23.46) |
| Mean symptom onset to hospital arrival (hrs) | | 45.90 | 24.2 |
| Time from admission to CT scan (hrs) | | 1.9 | 2.7 |
| Time from diagnosis to start of treatment (hrs) | | 6.36 | 3.64 |
| Mean Systolic blood pressure at admission | | 150 | 148 |
| Mean Diastolic blood pressure at admission | | 91.53 | 88.30 |
| Admission Mean Temperature | | 36.50 | 36.60 |

*(Continued)*

**Table 2.** (Continued)

| Variable | | Dead at 28 days N (%) | Alive at 28 days N (%) |
|---|---|---|---|
| Admission Mean Pulse rate | | 87.46 | 82.44 |
| Admission mean Glasgow coma scale score | | 7.6 | 12.18 |
| Admission mean modified ranking scale score | | 4.73 | 4.17 |
| Admission mean National Institute of Health Stroke Scale score | | 22.4 | 14.78 |

SBP = Systolic Blood Pressure, DBP = Diastolic Blood Pressure, PR = Pulse Rate, mRS = Modified Ranking Scale, NIHSS = National Institute of Health Stroke Scale, GCS = Glasgow Coma Scale, T° = Temperature

The results from Gompertz parametric baseline hazard distribution revealed that the 28-days case fatality rate of female patients was lower by 73% (AHR = 0.27, 95% CI:0.18–0.42) as compared to male patients. Similarly, the risk of death among patients with no family history of chronic disease was lower by 50% (AHR = 0.50, 95% CI:0.29-.87) compared to their counterparts. The estimated hazard of death among stroke patients with good GCS score was lower by 79% as compared to stroke patients with poor GCS score (AHR = 0.21, 95% CI:0.9–0.50). Moreover, stroke patients with no complications were 84% less likely to die than stroke patients with complications (AHR = 0.16, 95% CI:0.08–0.29).

Compared to ischemic stroke the hazard of fatality was higher by 38% in hemorrhagic stroke patients (AHR = 1.38, 95% CI:1.04–3.19). A one-hour delay from symptom onset to hospital arrival leads to a 49% increased risk of death (AHR = 1.49, 95% CI:1.57–2 .71). As well, one hour delay in time from confirmation of the diagnosis to treatment the hazard of death was higher by 3% (AHR = 1.03, 95% CI: 1.01–1.04). 28 days case fatality rate of admitted stroke patients with a respiratory rate greater than twenty was 7.26 times higher than patients with less than or equal to twenty respiratory rates (AHR = 7.26, 95% CI: 2.80–18.8). Besides, the hazard of death stroke among patients with one score increase in NIHSS score resulted in a 16% higher risk of death compared to one NIHSS score lower patients (AHR = 1.16, 95% CI:1.10–1.23) (Table 3). Survival probability curves derived from Log-rank Kaplan Meier of 28 days case fatality rate with different factors were shown (Figs 1–3).

## Discussion

### Main findings

This is the first study reporting case fatality rate for acute ischaemic and haemorrhagic stroke in Northwest Ethiopia. After twenty-eight days follow up greater than one-fourth of stroke patients have died. The results from Gompertz parametric baseline hazard distribution revealed that female sex, absence of a family history of chronic disease, good GCS score, and the absence of complications during hospital admission were factors that decrease the hazard of 28 days case fatality rate. While, hemorrhagic stroke sub-type, time from symptom onset to hospital arrival, time from confirmation of the diagnosis to initiation of treatment, a respiratory rate greater than 20, and an increase in NIHSS score were factors increasing the hazard of 28 days case fatality rate.

### Comparison with literatures

This prospective cohort study was conducted among stroke patients in Northwest Ethiopia to identify the 28 days follow-up case fatality rate and its determinants among admitted stroke patient. On a 28-day follow-up, 27.08% of patients have died which was comparable with a studies from Ethiopia (29.3%) (30.1%), and Uganda (26.8%) (15,19,20). However, the rate was

**Table 3. Determinants of 28 days case fatality rate among admitted stroke patients in public referral hospitals in Northwest Ethiopia, 2020/21.**

| Variables | Stroke status | | Crude hazard rate (95% CI) | Adjusted hazard rate (95% CI) |
|---|---|---|---|---|
| | Dead | Alive | | |
| **Sex** | | | | |
| Male | 80 | 179 | 1 | 1 |
| Female | 70 | 225 | 0.71(0.51–0.97) | 0. 27 (0.18–0.42) |
| **Age** | | | | |
| ≤ 60 | 90 | 191 | 1 | 1 |
| > 60 | 60 | 213 | 1.61 (1.38–1.84) | 1.37 (0.84–2.22) |
| **Education level** | | | | |
| Primary and below | 140 | 355 | 1 | 1 |
| Secondar and above | 10 | 49 | 0.55(0.32–0.94) | 1.10 (0.50–2.45) |
| **Alcohol usage** | | | | |
| None drinker | 70 | 235 | 1 | 1 |
| Drinker | 80 | 169 | 1.46(1.06–2.02) | 1.31 (0.67–1.48) |
| **Physical activity** | | | | |
| ≥ moderately active | 20 | 95 | 1 | 1 |
| ≤ sedentarily active | 130 | 309 | 1.81 (1.13–2.90) | 1.37 (0.68–2.74) |
| **Family history of chronic disease** | | | | |
| Yes | 36 | 25 | 1 | 1 |
| No | 114 | 379 | 0.30 (0.21–0.44) | 0.50 (0.29–0.87) |
| **Presence of comorbidity** | | | | |
| No | 65 | 219 | 1 | 1 |
| Yes | 85 | 185 | 1.54 (1.11–2.12) | 1.58 (0.45–2.16) |
| **Presence of complication** | | | | |
| Yes | 130 | 95 | 1 | 1 |
| No | 20 | 309 | 0.07(.04–0.11) | 0.16 (0.08–0.29) |
| **Type of stroke** | | | | |
| Ischemic | 60 | 274 | 1 | 1 |
| Haemorrhagic | 90 | 130 | 2.60 (1.88–3.61) | 1.38 (1.04–3.19) |
| **Time to hospital arrival** | 150 | 404 | 1.99 (1.37–2.93) | 1.49 (1.57–2 .71) |
| **Time to treatment** | 150 | 404 | 1.02 (1.01–1.03) | 1.03 (1.01–1.04) |
| **Admission Glasgow coma scale score** | | | | |
| Poor | 90 | 30 | 1 | 1 |
| Moderate | 50 | 165 | 0.19 (0.13–0.27) | 1.02(0.55–1.90) |
| Good | 10 | 209 | 0.03 (0.01–0.06) | 0.21(0.09–0.50) |
| **Diastolic blood pressure** | 150 | 404 | 1.01(1.003–1.02) | 1.00 (0.99–1.01) |
| **Respiratory rate** | | | | |
| ≤ 20 | 130 | 394 | 1 | 1 |
| > 20 | 20 | 10 | 4.18 (2.61–6.70) | 7.21(3.48–14.9) |
| **National Institute of Health Stroke Scale score** | 150 | 404 | 1.2 (1.17–1.24) | 1.16 (1.10–1.23) |

higher compared to studies from Nigeria (21.2%), (17.7%), Cameroon (23.2%), Kenya (18.4%), Brazil (12.5%), and Saudi (9.7%) [5, 7–10]. On contrary, the fatality rate was lower than in a study conducted in Tanzania (33.3%) and Senegal (38%) [9, 11]. The possible variation in mortality rate might be with the milder cases might not seek health care while the most severe case may die before reaching hospitals, set up of the hospital, complications, available resources, and comorbidities.

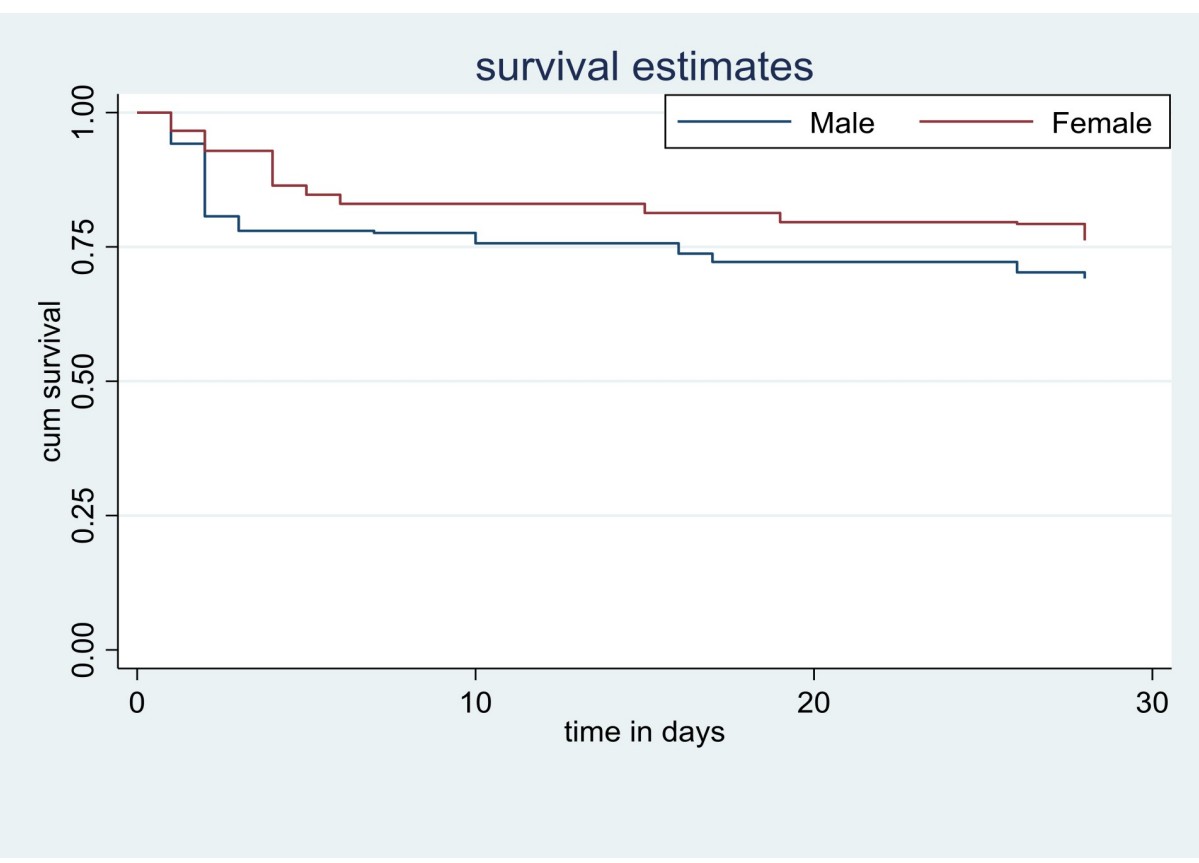

**Fig 1. Survival probability curves derived from Log rank Kaplan Meier of fatality after 28 day follow up and sex.**

Consistent with previous studies [5, 10, 12–15] the present study revealed that the 28 days case fatality rate was higher in hemorrhagic stroke patients compared to ischemic stroke patients.

This study found out the 28-days case fatality rate of female stroke patients was lower as compared to male stroke patients which were supported by a study from Nigeria [16]. This finding contradicted a study from Kenya [10]. The possible justification might be variation with behavioral risk factors like alcohol drinking and cigarette smoking [8, 17].

This study evidenced that the risk of death among patients with no family history of chronic disease was lower compared to their counterparts. Moreover, stroke patients without complications were less likely to die than stroke patients with complications. This finding is in line with existing literature [7–9, 12, 18–20].

If stroke patients were one hour delayed from symptom onset to hospital arrival, they were more likely to die. This finding is also in line with a study from the Democratic Republic of Congo [14]. While a study from Saudi reported time to hospital arrival was insignificant [8]. Possibly, time will be great in the prevention of either neurologic or medical complications and the other is in the prevention of ischemic to hemorrhagic transition.

The estimated hazard of death among stroke patients with a good GCS was lower as compared to stroke patients with a poor GCS. Similarly studies witnessed that a Poor GCS is a risk factor for mortality of admitted stroke patients [7, 12, 21, 22]. The explanation might be poor GCS indicates the severity of the brain injury, the presence of neurological complications like increased ICP and herniation which are fatal even if the patient survives it will leave devastating disability.

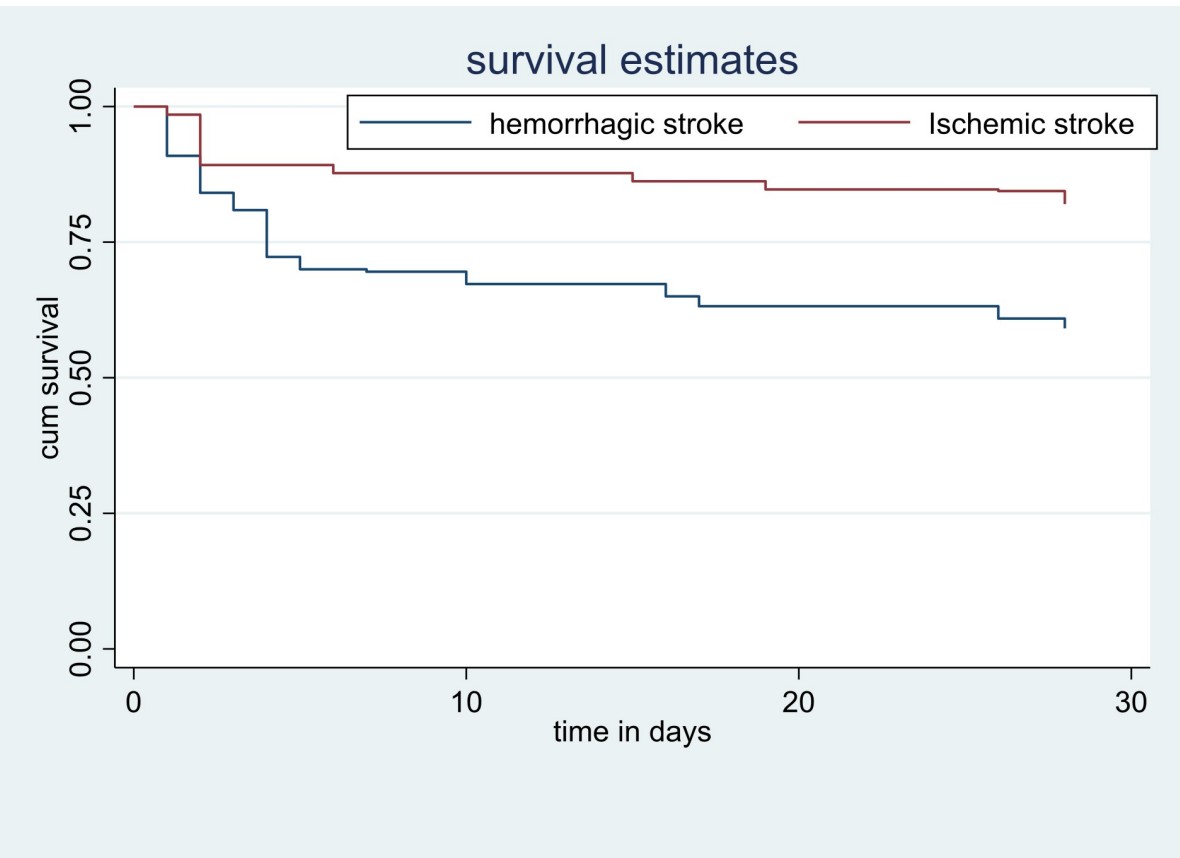

**Fig 2. Survival probability curves derived from Log rank Kaplan Meier of fatality after 28 day follow up and type of stroke.**

Respiratory rate was a significant predictor of case fatality rate among admitted stroke patients. Respiratory rates greater than twenty were seven times higher risk of death compared to their counterparts. Different studies supported the finding of the current study [12, 23]. The reason for this association might be tachypneic patients may have complications like aspiration pneumonia, pulmonary embolism, CHF, and other clinical conditions making the treatment difficult and the treatment outcome worrisome.

Besides, the hazard of death among stroke patients with higher NIHSS resulted in a higher risk of death compared to the ones with lower NIHSS scores which was in line with existing literature [5, 7, 12]. Possibly, it was prudent the NIHSS is a well-validated tool for assessing initial stroke severity irrespective of the subtype and has previously been shown to be associated with predicting mortality [24].

## Strength and limitations of the study

Important strengths of our study were, that it was a multicentered, prospective cohort study and the enrollment of CT scan confirmed stroke cases which make the fatality rate more accurate and reliable. The study provides a preliminary database on case fatality rate which can inform stroke management strategies and interventions required to decrease the fatality rate associated with stroke. Additionally, no patients were lost to follow-up. Our study included patients who were admitted to the hospital whereby stroke severity extremes might not be included. In addition, we excluded patients who died before having a CT scan case fatality rate might be underestimated.

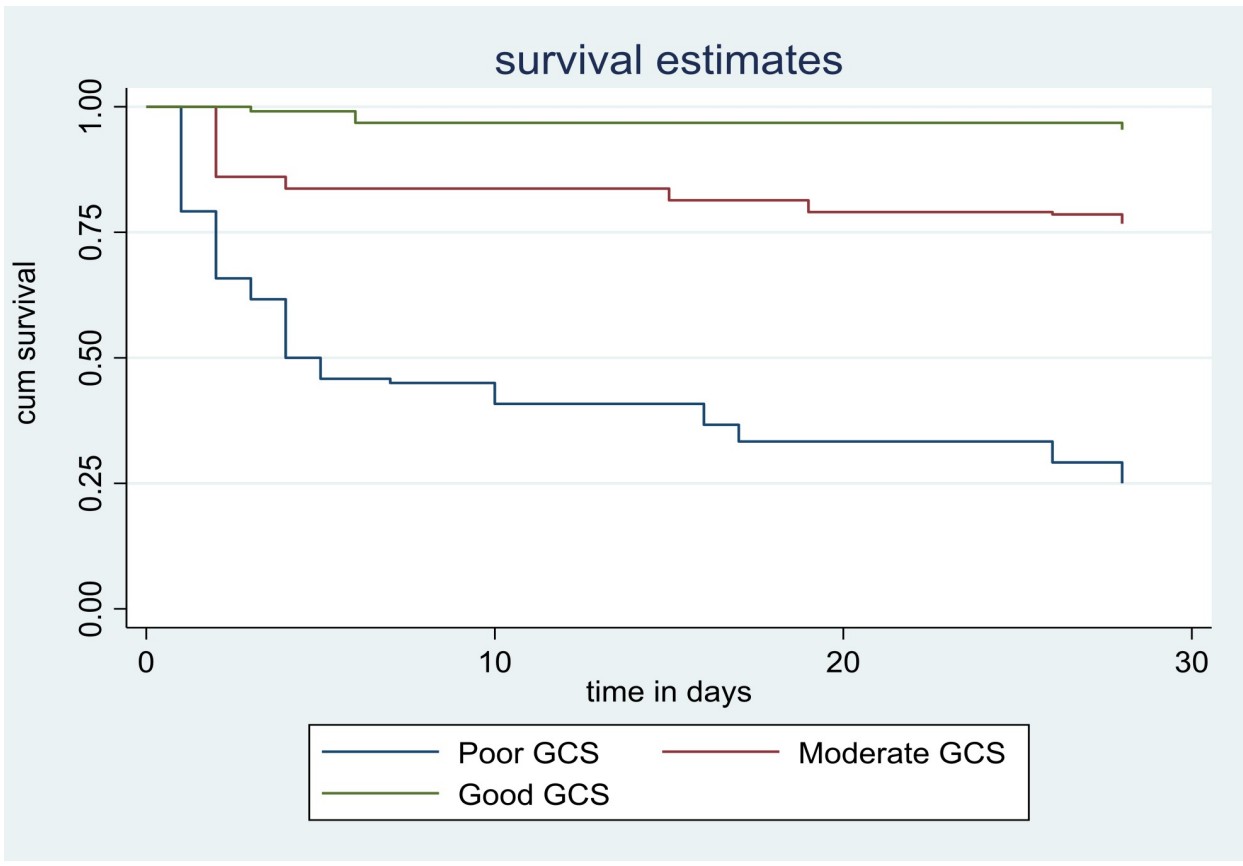

**Fig 3. Survival probability curves derived from Log rank Kaplan Meier of fatality after 28 day follow up and GCS score.**

We followed discharged patients by telephone, not by face-to-face interview. Thus, the detailed data on stroke severity, recovery, and disability could not be collected in this study and the accuracy of this self-reported event may limit the result.

## Conclusion

After a 28-days follow-up, greater than one-fourth of the patients have died. Female sex, absence of a family history of chronic disease, good GCS score, and the absence of complications during hospital admission were factors that decrease the hazard of 28 days case fatality rate. While, hemorrhagic stroke sub-type, time from symptom onset to hospital arrival, time from confirmation of the diagnosis to initiation of treatment, a respiratory rate greater than 20, and an increase in NIHSS score were factors increasing the hazard of 28 days case fatality rate. Therefore, preventive measures for individuals with a family history of chronic illness and comorbidities should be planned. Efforts should be made to establish best practices for acute stroke care in our settings. The establishment of separate stroke centers and a network of local and regional stroke centers with expertise in early stroke evaluation and management may address the challenges. Moreover, Prospective community-based stroke incidence and prevalence studies are required to define the true mortality of stroke in Ethiopia. Hence, future studies collecting community-based data, and carrying out a pooled data analysis may detect further determinants of stroke mortality rate.

## Supporting information

**S1 Data.**
(DTA)

## Acknowledgments

First and foremost we would like to acknowledge the participants of the study. Our special thanks and appreciation also goes to data collectors and staff of the University of Gondar Teaching Hospital, Tibebe Gion comprehensive specialized hospital, and Felege Hiwot referral hospital.

## Author Contributions

**Conceptualization:** Gashaw Walle Ayehu, Getachew Yideg Yitbarek, Tadeg Jemere, Edgeit Zewde.

**Data curation:** Gashaw Walle Ayehu, Getachew Yideg Yitbarek, Sisay Degno.

**Formal analysis:** Gashaw Walle Ayehu, Getachew Yideg Yitbarek, Sofonias Abebaw, Edgeit Zewde, Sisay Degno.

**Funding acquisition:** Gashaw Walle Ayehu.

**Investigation:** Gashaw Walle Ayehu, Ermias Sisay Chanie, Sofonias Abebaw, Ayele Mamo.

**Methodology:** Gashaw Walle Ayehu, Tadeg Jemere, Ermias Sisay Chanie, Sofonias Abebaw, Daniel Atlaw, Ayele Mamo, Melkalem Mamuye Azanaw.

**Project administration:** Gashaw Walle Ayehu, Ermias Sisay Chanie, Daniel Atlaw.

**Resources:** Gashaw Walle Ayehu, Daniel Atlaw, Assefa Agegnehu.

**Software:** Gashaw Walle Ayehu, Ermias Sisay Chanie, Dejen Getaneh Feleke, Sofonias Abebaw, Melkalem Mamuye Azanaw.

**Supervision:** Gashaw Walle Ayehu.

**Validation:** Gashaw Walle Ayehu, Dejen Getaneh Feleke, Daniel Atlaw, Melkalem Mamuye Azanaw.

**Visualization:** Gashaw Walle Ayehu, Assefa Agegnehu, Ayele Mamo.

**Writing – original draft:** Gashaw Walle Ayehu, Getachew Yideg Yitbarek, Tadeg Jemere, Ermias Sisay Chanie, Dejen Getaneh Feleke, Sofonias Abebaw, Edgeit Zewde, Assefa Agegnehu, Sisay Degno, Melkalem Mamuye Azanaw.

**Writing – review & editing:** Gashaw Walle Ayehu, Getachew Yideg Yitbarek, Tadeg Jemere, Dejen Getaneh Feleke, Edgeit Zewde, Assefa Agegnehu, Ayele Mamo, Sisay Degno, Melkalem Mamuye Azanaw.

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
