## [Decision Letter · Decision Letter 0]

10 Mar 2022

PONE-D-22-03279TWENTY EIGHT DAY MORTALITY RATE AND ITS DETERMINANTS AMONG ADMITTED STROKE PATIENTS IN PUBLIC REFERRAL HOSPITALS, NORTHWEST, ETHIOPIA: A PROSPECTIVE COHORT STUDYPLOS ONE

Dear Dr. Ayehu,

Thank you for submitting your manuscript to PLOS ONE. After careful consideration, we feel that it has merit but does not fully meet PLOS ONE’s publication criteria as it currently stands. Therefore, we invite you to submit a revised version of the manuscript that addresses the points raised during the review process.Please ensure that your decision is justified on PLOS ONE’s publication criteria and not, for example, on novelty or perceived impact. Specifically although it was concluded that the data provided in the study are valuable, reservations about their choice of statistical methods were raised. I invite you to submit a revised manuscript that accounts for the concerns raised by the reviewer. Please submit your revised manuscript by Apr 24 2022 11:59PM. If you will need more time than this to complete your revisions, please reply to this message or contact the journal office at plosone@plos.org. Please include the following items when submitting your revised manuscript:A rebuttal letter that responds to each point raised by the academic editor and reviewer(s). You should upload this letter as a separate file labeled 'Response to Reviewers'.A marked-up copy of your manuscript that highlights changes made to the original version. You should upload this as a separate file labeled 'Revised Manuscript with Track Changes'.An unmarked version of your revised paper without tracked changes. You should upload this as a separate file labeled 'Manuscript'.

We look forward to receiving your revised manuscript.

Kind regards,

Colin Johnson, Ph.D.

Academic Editor

PLOS ONE

Journal Requirements:

 [the funders had no role in study design, data collection and analysis, decision to publish, or preparation of the manuscript.]

[First and foremost we would like to acknowledge participants of the study. We would like to express our deepest gratitude to Debre Tabor University, for funding this research. Our special thanks and appreciation also goes to data collectors and staff of the University of Gondar Teaching Hospital, Tibebe Gion comprehensive specialized hospital, and Felege Hiwot referral hospital.]

 [the funders had no role in study design, data collection and analysis, decision to publish, or preparation of the manuscript.]

Reviewers' comments:

Reviewer's Responses to Questions

**Comments to the Author**

1. Is the manuscript technically sound, and do the data support the conclusions?

Reviewer #1: Partly

2. Has the statistical analysis been performed appropriately and rigorously? 

Reviewer #1: No

3. Have the authors made all data underlying the findings in their manuscript fully available?

Reviewer #1: No

4. Is the manuscript presented in an intelligible fashion and written in standard English?

Reviewer #1: No

5. Review Comments to the Author

Reviewer #1: The study by Gashaw Walle Ayehu and colleagues reported on the 28-day mortality rate following a hospital admission for stroke in three referral hospitals in Ethiopia, and analysed a number of risk factors associated with the increased or decreased mortality rate. The reported results are important as there is limited data on stroke outcomes in Ethiopia. The descriptive data provided in the study are valuable, but I have reservations about their choice of statistical methods. Moreover, I find the quality of reported results suboptimal. Some of the reported findings, including a dramatic excess death among ischaemic stroke patients compared to haemorrhagic, which was left without an explanation, is making me hesitant about recommending this manuscript for publication. I have provided an extensive list of comments, which hopefully can help the authors to improve their manuscript. I will leave the final decision with the Editor.

Abstract

Background: a missed word between “was” and “28”,

Results: rewrite this section and organise your findings more efficiently. Perhaps, group the risk factors/determinants into those that increase and those that decrease the 28-day mortality rate. List and properly name the risk factors, so that the readers can understand which sex was the predictor. For example, male sex.

Line 14. Please change the wording. The majority (80%) died in hospital and the rest after a discharge from hospital.

Introduction

Line 2. Please, change the wording. By 2020, stroke has become the second leading cause of death.

Please, check the reference N4, African continent is not responsible for 86% of all stroke deaths. It is an overestimation. I believe the correct wording would be that 86% of admitted stroke patients in Africa, die.

Inclusion and Exclusion criteria

p 12, line 2. Delete admitted patients, keep “All patients diagnosed with stroke and admitted …”

p12, line 3. Change the beginning of the new sentence: We excluded individuals who died before …

The exclusion criteria were specified to exclude people admitted with stroke who died before having a CT scan. These were likely the most severe stroke patients. If excluded from analysis, the mortality rates would be underestimated. Please, report how many patients with suspected stroke died on admission before they could undergo a CT scan. Address this in the limitation section. If the proportion is high (15-20%) among all stroke patients who died in hospital, you might want to run a sensitivity analysis adding these individuals.

p 12, line 13, a comma is missing between education level and alcohol use.

In the method section: please describe in detail how the 28 mortality rate was calculated, what was used in the numerator and denominator. Why 28 mortality rate? Should it be 28-day case fatality?

Definitions, no definition was provided for how the 28-day mortality rate was calculated, and what constituted the numerator and denominator. Therefore, I cannot clearly understand what rate was reported, however I wondered if a more appropriate terminology to use here would be 28-day case fatality. From what I gathered in the paper, the study included all stroke patients and the outcome of interest was death at 28 days from any cause, the rate was calculated as percentage, so it is a case fatality. How did you differentiate between first stroke and readmission for stroke? Did you include only the first hospital admission for stroke into the denominator or any subsequent hospital admission?

Study variables

Please, describe how crude hazard rate and adjusted hazard rate were calculated. How was the adjustment made?

Data collection

p 13, line 27. I was surprised to read that the data collection was carried out by a GP (general practitioner). Is this correct or was it just a translation issue? Do you have neurologists who review patients with stroke? If this information is indeed correct, you might want to say something more general, for example, “the data collection was carried out by a treating doctor and validated by another physician”, or something similar. This will get you around additional criticism. However, I do have my own reservations about the quality of stroke diagnosis in this study if there was no input from a neurologist or trained stroke physician with data collection. I read in a sentence describing the data quality that a neurologist has assessed the clarity and comprehensiveness of data collected, but how about the quality and accuracy of stroke diagnosis? Have they validated or assessed it in any way?

Table 1. Not the most informative and conventional presentation with these three columns, dead at 28 days and alive at 28 days, and then one for the total population. I would suggest to make a new table 1 with just the descriptive statistics separately for men and women admitted with stroke. This will give a more detailed information about your study population. Then, make table 2 with sociodemographic and other characteristics of men and women who died from stroke at 28 days only. Ideally, you would separate ischaemic and haemorrhagic stroke, as these two types have important differences in risk factors, and particularly, management. I would stress out the importance of presenting the results for specific stroke types, especially given the findings of much higher mortality in patients with ischaemic stroke reported in this study.

P 17, line 1. In brackets missing 0.87

P 17. Line 4. The authors reported that mortality was 82% higher among ischaemic stroke patients compared to haemorrhagic. This is an odd finding. There is a general agreement in medicine that haemorrhagic stroke is associated with higher mortality than ischaemic. This is observed in all populations despite their ethnic origins. This is probably the first study to report such dramatic excess in ischaemic stroke death. The possible explanation for the increased mortality rate here is some kind of a misclassification. Perhaps, the type of stroke was not well defined in this study, possibly due to a low quality of the brain imaging, or maybe because of the large proportion of individuals with stroke type not specified, which was actually haemorrhagic stroke. Although, usually cases of unspecified stroke are ischaemic strokes. The authors have mentioned that the stroke diagnosis was confirmed by a CT scan. Do they have access to the MRI or angio? Were all hospitals equipped with the same medical devices? Alternative explanation, might be that patients with severe haemorrhagic stroke don’t get admitted to the hospitals included in the study and were transported to a different facility. I really find it difficult to believe a report that describes such a dramatic difference, unless they can offer a convincing explanation. The current manuscript just states the findings on page 19 without any explanation. However, when I checked the results, in table 2 which reports Hazard Ratios, authors have chosen haemorrhagic stroke as a reference category, and the results showed a lower HR for ischaemic stroke 0.38 (0.2- 0.53). so I am puzzled about why they reported 82%.

Page 17, line 6. Instead of “As the Time”, use “If” or “A one hour delay lead to..”

The graphs with Kaplan-Mayer survival curves are not numbered and not referred to in the results section. They are just pinned at the end of the text. Please, name the graphs and refer to them in the text.

In the conclusion paragraph, the opening sentence brings up the chronic and infectious disease, but these have not been assessed as comorbidities in the analysis. I would suggest omit the first sentence. Conclusion should be rewritten to provide the summary of the findings, put the findings in the broader context, as well as discuss the implications for public health, research and policy. Next, the authors have discussed the importance of establishing stroke centres and prospective community-based stroke studies, please, omit or explain how these are relevant to the study findings.

At the end of the paper, authors have provided a list of acronyms and abbreviations, but did not use some of them in the manuscript. Please, revise the list and delete unnecessary, including HIV, AF, HIV and some other. Second, it would be really interesting to know if stroke outcomes, either survival curves or 28-day mortality rate, were different between HIV positive and HIV negative patients. If the HIV status was known, please, add there results to the text.

6. PLOS authors have the option to publish the peer review history of their article (what does this mean?). If published, this will include your full peer review and any attached files.

Reviewer #1: **Yes: **Olena Seminog

---

## [Author Response · Author response to Decision Letter 0]

19 Apr 2022

Dear respected Chief editor and reviewers I have tried to go through all the comments and suggestions, tried to make our manuscript more clear. specifically we have sent the data as supplementary file and it can be accessed through your data base. thank you indvance

---

## [Decision Letter · Decision Letter 1]

26 May 2022

PONE-D-22-03279R1TWENTY-EIGHT DAYS STROKE FATALITY RATE AND ITS DETERMINANTS AMONG ADMITTED STROKE PATIENTS IN PUBLIC REFERRAL HOSPITALS, NORTHWEST, ETHIOPIA: A PROSPECTIVE COHORT STUDYPLOS ONE

Dear Dr. Ayehu,

Thank you for submitting your manuscript to PLOS ONE. After careful consideration, we feel that it has merit but does not fully meet PLOS ONE’s publication criteria as it currently stands. Therefore, we invite you to submit a revised version of the manuscript that addresses the points raised during the review process.

We look forward to receiving your revised manuscript.

Kind regards,

Colin Johnson, Ph.D.

Academic Editor

PLOS ONE

Reviewers' comments:

Reviewer's Responses to Questions

**Comments to the Author**

1. If the authors have adequately addressed your comments raised in a previous round of review and you feel that this manuscript is now acceptable for publication, you may indicate that here to bypass the “Comments to the Author” section, enter your conflict of interest statement in the “Confidential to Editor” section, and submit your "Accept" recommendation.

Reviewer #1: (No Response)

2. Is the manuscript technically sound, and do the data support the conclusions?

Reviewer #1: Partly

3. Has the statistical analysis been performed appropriately and rigorously? 

Reviewer #1: No

4. Have the authors made all data underlying the findings in their manuscript fully available?

Reviewer #1: Yes

5. Is the manuscript presented in an intelligible fashion and written in standard English?

Reviewer #1: Yes

6. Review Comments to the Author

Reviewer #1: Reviewer’s comments

Terminology. Stroke fatality rate is wrong.

Dear Editor,

The authors have made important changes to the manuscript, making it much stronger. However, they still didn’t get the terminology quite right. 1. They have changed 28 mortality rate to stroke fatality rate, which is a wrong term. In their response letter, they have agreed that the reported rate is case fatality rate. However, they are not using the correct terminology in the manuscript. It is important that the authors change the terminology. Since, fatality rate is the death rate observed in a designated series of individuals affected by a simultaneous event (e.g. victims of a disaster). This term should be avoided. – Dictionary of Epidemiology by M. Porta. Instead, use case fatality or case-fatality rate. Some researchers disagree about “rate”, because from a mathematical point of view, by taking values in the range 0 to 100%, it is not a rate, but risk. However, case-fatality rate remains widely used. Check this recent paper:

Hatem A. Wafa ,Charles D. A. Wolfe,Ajay Bhalla,Yanzhong Wang

Long-term trends in death and dependence after ischaemic strokes: A retrospective cohort study using the South London Stroke Register (SLSR).

2. Next, some of the findings presented are hard to believe. In particular, that ischaemic stroke had a higher hazard rate of 28-day case fatality than haemorrhagic stroke after adjusting (Table 2). I would strongly suggest that the authors check their data and their model again.

In the initial response, they have acknowledged that a mistake occurred with data transferring, which resulted in a wrong conclusion. It is likely that something like this has happened again.

Further comments to authors

Dear Author, thank you for revising the manuscript, which made it look much stronger. However, I am not completely satisfied with the quality of your paper and would like you to work on it again. These are my comments.

3. Abstract and text – Abbreviations - AHR and CHR. Please, when you introduce these words, first write them in full with the abbreviations in brackets. You can then use the abbreviations. In Table 2 headings, use the words not abbreviations.

4. Line 6, page 34. You are making a misleading statement about 86% in-hospital mortality. And using a wrong reference again. Remove reference 2 about stroke and hyperthermia. It doesn’t give information about stroke statistics. Change your statement that Africa accounts for 86% in-hospital death, because it is wrong. I have made this comment before.

5. Line 19, page 34. The sentence needs a reference to support it. The decreased percentage of stroke hospitalization and mortality in developed countries likely reflects the advancements in acute stroke care.

You can use the following as a citation

Seminog OO, Scarborough P, Wright FL, Rayner M, Goldacre MJ. Determinants of the decline in mortality from acute stroke in England: linked national database study of 795 869 adults. BMJ. 2019 May 22;365:l1778. doi: 10.1136/bmj.l1778.

6. Line 4, page 35 Remove Study aim. Replace with Study design and settings

7. Line 11, page 35. Formatting. Remove a full stop

8. Line 12 Remove source and sample size. Replace with Study population.

9. Line 15. Remove the number in brackets (554)

10. Line 16. You don’t need to say that you used the convenient sampling method.

11. Line 10, page 36. Change the dependent variable to 28-days case-fatality rate

Type of stroke

12. Table 1 type of stroke and Line 6, page 39. Check your data again. There is another error somewhere in your analysis or elsewhere. Around 60% of strokes in your study were classified as haemorrhagic and around 40% as ischaemic. It is very unlikely that you have more haemorrhagic strokes than ischaemic in Ethiopia. Change the sentence “Haemorrhagic stroke was the most prevalent…”, as it is not true. Haemorrhagic stroke is not more common or prevalent in any country. If the results remain the same after you have checked the data and analysis, you must discuss in study limitations why there were more haemorrhagic strokes in your study. Are ischaemic strokes too mild for the admission threshold? Do ischaemic stroke patients go to another hospital for some reason? These are just some possibilities for you to think about.

13. Table 2. Describe in methods how you did the adjustment of hazard rate? What were your confounders? The adjustment made a big difference to HR by stroke type. The findings that HR for ischaemic stroke was higher than for haemorrhagic are hard to believe. Check your model.

14. Line 4, page 42. Change the wording. For example, “We reported 150 deaths from stroke at 28 days among 554 patients hospitalised with stroke. There were 90 deaths were from haemorrhagic and 60 from ischaemic stroke. Next, report CFR for ischaemic and haemorrhagic stroke in men and women separately. Check your data really well. When reporting your findings, do not confuse the proportion of ischaemic and haemorrhagic strokes in admitted patients, and the proportion of these two types in patients who died. The classic scenario would be - more ischaemic strokes on admission (higher %), but more deaths from haemorrhagic stroke at 28 days. So far, your presentation of results is confusing.

15. Line 7, page 42. What is the “follow-up stroke fatality rate”? Change this term, as it is not something recognised in epidemiology. Are you talking about death in the community after patient was discharged?

16. Line 9 to 13, page 42. Remove this paragraph. It doesn’t provide any useful information, as you are only listing that the variables that you’ve put in the model.

17. Table 1. Missing percentages and numbers in numerous places (married female, missing number in the column divorced female, missing % in non-drinker female and missing number in past drinker, etc)

18. Table 2 and Table 3. Authors have changed the Table 1 to present findings for men and women separately. They should do the same for Tables 2 and 3. This is particularly important, since female sex seems to be protective.

Discussion

19. In the first paragraph of discussion, summarise the main findings. Then, discuss these findings and compare with literature.

20. Line 4. Remove the first sentence and rephrase the second sentence. You could say this “This is the first study (check if this is correct) reporting case fatality rate for acute ischaemic and haemorrhagic stroke in Northwest Ethiopia. We found that nearly one third of stroke patients dies within 28 days after hospital admission. Case fatality was higher for haemorrhagic stroke (or not)

21. I would advise the authors to use the following subheadings in the discussion. Main findings. Comparison with literature. Strengths and limitations. Conclusions

22. Line 13, page 45. The authors report that 28 day mortality was higher in ischaemic stroke. Not only this sounds wrong, but also contradict their own findings reported earlier, line 4 and line 19, page 42. You reported a 38% higher hazard rate for haemorrhagic stroke. Check your data.

Conclusions

23. Line 13, page 46. Instead of listing the variables that you’ve used in the analysis, provide the specific answers. For example, don’t say type of stroke, but haemorrhagic stroke, not sex, but male sex, and so on. This whole sentence might work better in the discussion in main findings section. But I will leave it with you to decide.

24. Remove acronyms at the end.

7. PLOS authors have the option to publish the peer review history of their article (what does this mean?). If published, this will include your full peer review and any attached files.

Reviewer #1: No

---

## [Author Response · Author response to Decision Letter 1]

23 Jun 2022

Thank you Dear editor and Reviewer, we have made all the changes and adjustments as requested/commented. in addition we have given justifications for the questions raised in author response form. Thank you !

---

## [Decision Letter · Decision Letter 2]

9 Aug 2022

PONE-D-22-03279R2CASE FATALITY RATE AND ITS DETERMINANTS AMONG ADMITTED STROKE PATIENTS IN PUBLIC REFERRAL HOSPITALS, NORTHWEST, ETHIOPIA: A PROSPECTIVE COHORT STUDY YPLOS ONE

Dear Dr. Ayehu,

Hello, Your manuscript has received a positive review and is recommended for publication with a few suggested minor changes to the grammar of the text. Please look over the suggested changes and make changes where needed. Upon submission of the revised manuscript, I anticipate acceptance without further review.

We look forward to receiving your revised manuscript.

Kind regards,

Colin Johnson, Ph.D.

Academic Editor

PLOS ONE

Journal Requirements:

Reviewers' comments:

Reviewer's Responses to Questions

**Comments to the Author**

1. If the authors have adequately addressed your comments raised in a previous round of review and you feel that this manuscript is now acceptable for publication, you may indicate that here to bypass the “Comments to the Author” section, enter your conflict of interest statement in the “Confidential to Editor” section, and submit your "Accept" recommendation.

Reviewer #1: All comments have been addressed

2. Is the manuscript technically sound, and do the data support the conclusions?

Reviewer #1: Yes

3. Has the statistical analysis been performed appropriately and rigorously? 

Reviewer #1: Yes

4. Have the authors made all data underlying the findings in their manuscript fully available?

Reviewer #1: No

5. Is the manuscript presented in an intelligible fashion and written in standard English?

Reviewer #1: Yes

6. Review Comments to the Author

Reviewer #1: Dear Editor,

The manuscript was revised to a high standard and can be published once the authors make a few minor changes suggested below.

Dear authors,

I would like to praise your perseverance and all the hard work you have put into preparing and revising this manuscript. This work has significantly improved the manuscript. Please, make a few more changes and good luck with publishing your study.

In the abstract, when you report the findings, please remove the colon symbol “:” when reporting the 95%CI.

Table 3, do not use the asterisk symbols *, **, or *** in the table, as they do not provide any valuable additional information. The 95% confidence intervals provided in the table give all the information that the readers require. Remove the footnotes explaining the asterisks as well.

Throughout the manuscript there are words that start with a capital where it should be a lower case. For example, line 9 p 33, neurology ward of the hospitals (Hospital starts with a capital): line 2 page 36, Internal Medicine Resident, it should be all capitals or all lower cases, but not some words. Please, check the manuscript for these typos before resubmitting the final version to the journal.

In the conclusion section, in the new paragraph that you have added, you have listed all the important determinants that increase and decrease stroke case fatality together. This can confuse the readers. Can you either separate these in two groups, those determinants that were associated with a higher 28 case fatality rate and those that were associated with a lower 28-day case fatality rate? This will make the final part of your paper more clear. Second option might be to only list the parameters that were associated with a lower 28 case fatality rate.

Please, correct a minor grammatical error in the last sentence, you can say “Hence, future studies collecting community-based data…"

7. PLOS authors have the option to publish the peer review history of their article (what does this mean?). If published, this will include your full peer review and any attached files.

Reviewer #1: **Yes: **Olena Seminog

---

## [Author Response · Author response to Decision Letter 2]

15 Aug 2022

Thank you Dear reviewer and editor, We have made the requested changes and accepted the comments

---

## [Editor Report · Decision Letter 3]

19 Aug 2022

CASE FATALITY RATE AND ITS DETERMINANTS AMONG ADMITTED STROKE PATIENTS IN PUBLIC REFERRAL HOSPITALS, NORTHWEST, ETHIOPIA: A PROSPECTIVE COHORT STUDY Y

PONE-D-22-03279R3

Dear Dr. Ayehu,

We’re pleased to inform you that your manuscript has been judged scientifically suitable for publication and will be formally accepted for publication once it meets all outstanding technical requirements.

Kind regards,

Colin Johnson, Ph.D.

Academic Editor

PLOS ONE
---

## [Editor Report · Acceptance letter]

26 Aug 2022

PONE-D-22-03279R3 

Case fatality rate and its determinants among admitted stroke patients in public referral hospitals, Northwest, Ethiopia: A prospective cohort study 

Dear Dr. Ayehu:

I'm pleased to inform you that your manuscript has been deemed suitable for publication in PLOS ONE. Congratulations! Your manuscript is now with our production department. 

Kind regards, 

on behalf of

Dr. Colin Johnson 

Academic Editor

PLOS ONE